# DITTO: Offline Imitation Learning with World Models

## Abstract

For imitation learning algorithms to scale to real-world challenges, they must handle high-dimensional observations, offline learning, and policy-induced covariate-shift. We propose DITTO, an offline imitation learning algorithm which addresses all three of these problems. DITTO optimizes a novel distance metric in the latent space of a learned world model: First, we train a world model on all available trajectory data, then, the imitation agent is unrolled from expert start states in the learned model, and penalized for its latent divergence from the expert dataset over multiple time steps. We optimize this multi-step latent divergence using standard reinforcement learning algorithms, which provably induces imitation learning, and empirically achieves state-of-the art performance and sample efficiency on a range of Atari environments from pixels, without any online environment access. We also adapt other standard imitation learning algorithms to the world model setting, and show that this considerably improves their performance. Our results show how creative use of world models can lead to a simple, robust, and highly-performant policy-learning framework.

## 1 Introduction

Imitation learning (IL) is an approach to policy learning which bypasses reward specification by directly mimicking the behavior of an expert demonstrator. The simplest kind of IL, behavior cloning (BC), trains an agent to predict an expert's actions from observations, then acts on these predictions at test time. However, this approach fails to account for the sequential nature of decision problems, since decisions at the current step affect which states are seen later, breaking the IID assumption of standard supervised learning algorithms. The distribution of states seen at test time will differ from those seen during training unless the expert training data covers the entire state space, or the agent makes no mistakes. This distribution mismatch, or covariate shift, leads to a compounding error problem: initially small prediction errors lead to small changes in state distribution, which lead to larger errors, and eventual departure from the training distribution altogether (Pomerleau, 1989). Intuitively, the agent has not learned how to act under its own induced distribution. This was formalized in the seminal work of Ross & Bagnell (2010), who gave a tight regret bound on the difference in return achieved by expert and learner, which is quadratic in the episode length for BC.

Follow-up work in Ross et al. (2011) showed that a linear bound on regret can be achieved if the agent learns online in an interactive setting with the expert: Since the agent is trained *under its own distribution* with expert corrections, there is no distribution mismatch at test-time. This works well when online learning is safe and expert supervision is possible, but is untenable in many real-world use-cases such as robotics, where online learning can be unsafe, time-consuming, or otherwise infeasible. This captures a major open challenge in imitation learning: on one hand, we want to generate data on-policy to avoid covariate shift, but on the other hand, we may not be able to learn online due to safety or other concerns. The algorithm we present, DITTO, solves this *offline imitation learning* challenge in a way that scales to high-dimensional observations such as pixels. To the best of our knowledge, we are the first to solve these Atari benchmarks in the imitation learning setting where we train completely offline, from pixels alone, and consistently recover expert performance.

Ha & Schmidhuber (2018) propose a two-stage approach to policy learning, where agents first learn to predict the environment dynamics with a recurrent neural network called a "world model" (WM), and then learn

the policy inside the WM alone. This approach is desirable since it enables on-policy learning *offline*, in the world model. Similar model-based learning methods have recently achieved success in the adjacent setting of online RL (Hafner et al., 2021), and impressive zero-shot transfer of policies trained solely in the WM to physical robots (Wu et al., 2022).

When learning a policy, it is important to quantify how well the policy generalizes to unseen inputs. However, in imitation learning there is a conceptual difficulty in measuring generalization performance: Although we could evaluate policy performance on held-out expert state-action pairs (e.g. by measuring prediction accuracy), this fails to reflect the performance we should expect from the policy at test-time, because we are not evaluating the policy in the state distribution it will be acting under, namely *its own induced distribution*. World models offer a solution to this problem: given a learned dynamics model, we can perform roll-outs with our learned policy *from expert starting states*, and measure the divergence in the latent space *over multiple time-steps*. This gives us an offline, on-policy divergence measure which captures the sequential nature of the imitation learning decision problem. This is the key insight underlying the strong empirical performance of DITTO, and forms the basis of our theoretical contribution. To confirm the generalization properties of our algorithm, we evaluate in environments for which there is an extrinsic reward function (which the imitation learner does not have access to), and study the relationship between our proposed divergence measure and the extrinsic reward. As shown in Figure 2, we find that off-policy metrics like expert action prediction accuracy are not predictive of final return in the environment, whereas our on-policy latent divergence measure is predictive of extrinsic return.

We combine the above insights to propose a new imitation learning algorithm called **Dream Imitation (DITTO)**. DITTO leverages learned world models in a novel way to induce reward-free imitation learning: first, expert trajectories are mapped into the model latent space; then, an imitation policy is unrolled from arbitrary expert start states in the model latent space. Finally, we directly compare the *on-policy* rollouts to the expert trajectories in the model latent space using a simple, fixed distance function defined in the latent space. Our key insight is that the world model latent space provides a natural trajectory divergence measure which we can do imitation learning over, without any intermediary learned reward model or discriminator. This enables robust imitation learning in a domain-agnostic way. Using this divergence measure, DITTO casts the offline imitation learning challenge as an online RL problem in a learned world model.

Our main contributions are summarized as follows:

- We introduce DITTO, a novel imitation learning framework which leverages learnt world models to cast offline imitation as online RL in the model latent space.

- We show that the latent space induced by dynamics learning provides a natural and generic divergence measure for policy learning. The latent distance function we demonstrate is simpler, more robust, and results in more performant policy learning compared to learned discriminators, such as those used in adversarial imitation learning.

- We demonstrate DITTO outperforms standard imitation learning baselines by a significant margin - to the best of our knowledge we are the first to consistently recover expert performance in the Atari benchmarks we study from pixels, completely offline.

- To provide a more comprehensive evaluation of the thus far underexplored area of offline imitation learning from pixels, we present two novel extensions of baseline IL algorithms (BC, GAIL) to the world model, offline setting (D-BC, D-GAIL).

## 2 Related Work

### 2.1 Imitation Learning

Imitation learning algorithms can be classified according to the set of resources needed to produce a good policy. Ross et al. (2011) give strong theoretical and empirical results in the online interactive setting, which assumes that we can both learn while acting online in the real environment, and that we can interactively

query an expert policy to e.g. provide the learner with the optimal action in the current state. Follow-up works have progressively relaxed the resource assumptions needed to produce good policies. Sasaki & Yamashina (2021) show that the optimal policy can be recovered with a modified form of BC when learning from imperfect demonstrations, given a constraint on the expert sub-optimality bound. Brantley et al. (2020) study covariate shift in the online, non-interactive setting, and demonstrate an approximately linear regret bound by jointly optimizing the BC objective with a novel policy ensemble uncertainty cost, which encourages the learner to return to and stay in the distribution of expert support. They achieve this by augmenting the BC objective with the following uncertainty cost term:

$$\text{Var}_{\pi \sim \Pi_E} \left( \pi(a|s) \right) = \frac{1}{E} \sum_{i=1}^{E} (\pi_i(a|s) - \frac{1}{E} \sum_{j=1}^{E} \pi_j(a|s))^2 \tag{1}$$

This term measures the total variance of a policy ensemble $\Pi_E = \{\pi_1, ..., \pi_E\}$ trained on disjoint subsets of the expert data. They optimize the combined BC plus uncertainty objective using standard online RL algorithms, and show that this mitigates covariate shift.

Inverse reinforcement learning (IRL) can achieve improved performance over BC by first learning a reward from the expert demonstrations for which the expert is optimal, then optimizing that reward with on-policy reinforcement learning. This two-step process, which includes on-policy RL in the second step, helps IRL methods mitigate covariate shift due to train and test distribution mismatches. However, the learned reward function can fail to generalize outside of the distribution of expert states which form its support.

One line of work treats IRL as *divergence minimization*: instead of directly copying the expert actions, they minimize a divergence measure between expert and learner state distributions

$$\min_{\pi} \mathbb{D} \left( \rho^\pi, \rho^E \right) \tag{2}$$

where $\rho^\pi(s, a) = (1 - \gamma) \sum_{t=0}^{\infty} \gamma^t P(s_t = s, a_t = a)$ is the discounted state-action distribution induced by $\pi$, and $\mathbb{D}$ is a divergence measure between probability distributions. The popular GAIL algorithm (Ho & Ermon, 2016) constructs a minimax game in the style of GANs (Goodfellow et al., 2014) between the learner policy $\pi$, and a discriminator $D_\psi$ which learns to distinguish between expert and learner state distributions

$$\max_{\pi} \min_{D_\psi} \mathbb{E}_{(s,a) \sim \rho^E} \left[ -\log D_\psi(s, a) \right] + \mathbb{E}_{(s,a) \sim \rho^\pi} \left[ -\log \left( 1 - D_\psi(s, a) \right) \right] \tag{3}$$

This formulation minimizes the Jensen-Shannon divergence between the expert and learner policies, and bounds the expected return difference between agent and expert. However, Wang et al. (2019) point out that adversarial reward learning is inherently unstable since the discriminator is always trained to penalize the learner state-action distribution, even if the learner has converged to the expert policy. This finding is consistent with earlier work (Brock et al., 2019) which observed discriminator overfitting, necessitating early stopping to prevent training collapse. Multiple works have reported failure getting GAIL to work with high-dimensional observations, such as those in the pixel-based environments we study (Brantley et al., 2020) (Reddy et al., 2020).

To combat problems with adversarial training, Wang et al. (2019) and Reddy et al. (2020) consider reducing IL to RL on an intrinsic reward

$$r(s, a) = \begin{cases} 1 & \text{if } (s, a) \in \mathcal{D}^E \\ 0 & \text{otherwise} \end{cases} \tag{4}$$

where $\mathcal{D}^E$ is the expert dataset. While this sparse formulation is impractical e.g. in continuous action settings, they show that a generalization of the intrinsic reward using support estimation by random network distillation (Burda et al., 2019) results in stable learning that matches the performance of GAIL without adversarial training. Ciosek (2022) showed that this formulation is equivalent to divergence minimization under the total variation distance, and produced a bound on the difference in extrinsic reward achieved between the expert and a learner trained with this approach.

## 2.2 World Models

World models have recently emerged as a promising approach to model-based learning. Ha & Schmidhuber (2018) defined the prototypical two-part model: a variational autoencoder (VAE) is trained to reconstruct observations from individual frames, while a recurrent state-space model (RSSM) is trained to predict the VAE encoding of the next observation, given the current latent state and action. World models can be used to train agents entirely inside the learned latent space, without the need for expensive decoding back to the observation space. Hafner et al. (2020) introduced Dreamer, an RL agent which is trained purely in the latent space of the WM, and successfully transfers to the true environment at test-time. Wu et al. (2022) showed that the same approach can be used to simultaneously learn a model and agent policy to control a physical quadrupedal robot online, without the control errors usually associated with transferring policies trained only in simulation to a physical system (Hwangbo et al., 2019).

In this work, we propose the use of world models to address a number of common problems in imitation learning. Intrinsic rewards which induce imitation learning, like those introduced in Reddy et al. (2020) and Wang et al. (2019), can pose challenging online learning problems, since the rewards are sparse or require tricky additional training procedures to work in high-dimensional observation spaces. Similarly, approaches like GAIL (Ho & Ermon, 2016) and AIRL (Fu et al., 2018) require adversarial on-policy training that is difficult to make work in practice. Rafailov et al. (2021) propose an approach similar to ours, which uses model-based rollouts to produce on-policy latent trajectories to train the policy. However, their reward model is learned via an adversarial objective, and so can in principle suffer from the same adversarial collapse issues mentioned above. In contrast, our approach remedies both the online learning and reward specification problems by performing on-policy learning offline, in the latent space of the world model, and uses a natural divergence measure as reward: distance between learner and expert in the world model latent space. This provides a conceptually simple and dense reward signal for imitation by reinforcement learning, which we find outperforms competitive approaches in data efficiency and asymptotic performance.

# 3 Dream Imitation

We study imitation learning in a partially observable Markov decision process (POMDP) with discrete time-steps and actions, and high dimensional observations generated by an unknown environment. The POMDP $\mathcal{M}$ is composed of the tuple $\mathcal{M} = (\mathcal{S}, \mathcal{A}, \mathcal{X}, \mathcal{R}, \mathcal{T}, \mathcal{U}, \gamma)$, where $s \in \mathcal{S}$ is the state space, $a \in \mathcal{A}$ is the action space, $x \in \mathcal{X}$ is the observation space, $\gamma$ is the discount factor, and $r = \mathcal{R}(s, a)$ is the reward function. The transition dynamics are Markovian, and given by $s_{t+1} \sim \mathcal{T}(\cdot \mid s_t, a_t)$. The agent does not have access to the underlying states, and only receives observations represented by $x_t \sim \mathcal{U}(\cdot \mid s)$. The goal is to maximize the discounted sum of extrinsic (environment) rewards $\mathbb{E}[\Sigma_t \gamma^t r_t]$, which the agent does not have access to.

Training proceeds in two parts: we first learn a world model from recorded sequences of observations, then train an actor-critic agent to imitate the expert in the world model. The latent dynamics of the world model define a fully observable Markov decision process (MDP), since the model states $\hat{s}_t$ are Markovian. Model-based rollouts always begin from an observation drawn from the expert demonstrations, and continue for a fixed set of time steps $H$, the agent training horizon. The agent is rewarded for matching the latent trajectory of the expert.

## 3.1 Preliminaries

We show that bounding the learner-expert state distribution divergence in the world model also bounds their return difference in the actual environment, and connect our method to the *IL as divergence minimization* framework (Ghasemipour et al., 2019). Rafailov et al. (2021) showed that for a learned dynamics model $\widehat{\mathcal{T}}$ whose total variation from the true transitions is bounded such that $\mathbb{D}_{\mathrm{TV}}(\mathcal{T}(s, a), \widehat{\mathcal{T}}(s, a)) \leq \alpha \quad \forall (s, a) \in \mathcal{S} \times \mathcal{A}$ and $R_{\max} = \max_{(s,a)} \mathcal{R}(s, a)$ then

$$\left| \mathcal{J}(\pi^E, \mathcal{M}) - \mathcal{J}(\pi, \mathcal{M}) \right| \leq \underbrace{\alpha \frac{R_{\max}}{(1 - \gamma)^2}}_{\text{learning error}} + \underbrace{\frac{R_{\max}}{1 - \gamma} \mathbb{D}_{\mathrm{TV}}\left( \rho_{\mathcal{M}}^E, \rho_{\widehat{\mathcal{M}}}^\pi \right)}_{\text{adaptation error}} \tag{5}$$

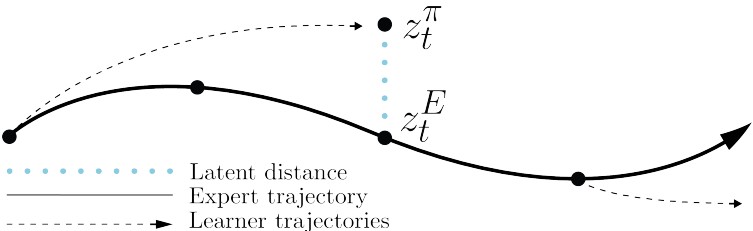

Latent distance
Expert trajectory
Learner trajectories

Figure 1: The learner begins from random expert latent states during training, and generates on-policy latent trajectories in the world model. The intrinsic reward 8 encourages the learner to recover from its mistakes over multiple time steps to match the expert trajectory.

where $\mathcal{J}(\pi, \mathcal{M})$ is the expected return of policy $\pi$ in MDP $\mathcal{M}$, and $\widehat{\mathcal{M}}$ is the "imagination MDP" induced by the world model. This implies the difference between the expert return and the learner return in the true environment is bounded by two terms, 1) a term proportional to the model approximation error $\alpha$, which could in principle be reduced with more data, and 2) a model domain adaptation error term, which captures the generalization error of a model trained under data from one policy, and deployed under another. Rafailov et al. (2021) also show that bounding the divergence between *latent* distributions upper bounds the true state distribution divergence. Formally, given a latent representation of the transition history $z_t = q(x_{\leq t}, a_{<t})$ and a belief distribution $P(s_t \mid x_{\leq t}, a_{<t}) = P(s_t \mid z_t)$, then if the policy conditions only on the latent representation $z_t$ such that the belief distribution is independent of the current action $P(s_t \mid z_t, a_t) = P(s_t \mid z_t)$, then the divergence between the latent state distribution of the expert and learner upper bounds the divergence between their true state distribution:

$$\mathbb{D}_f(\rho^\pi_\mathcal{M}(x,a) \parallel \rho^E_\mathcal{M}(x,a)) \leq \mathbb{D}_f(\rho^\pi_\mathcal{M}(s,a) \parallel \rho^E_\mathcal{M}(s,a)) \leq \mathbb{D}_f(\rho^\pi_\mathcal{M}(z,a) \parallel \rho^E_\mathcal{M}(z,a)) \quad (6)$$

Where $\mathbb{D}_f$ is a generic $f$-divergence , e.g. KL or TV. This result, along with equation 5, suggests that minimizing divergence in the model latent space is sufficient to bound the expected expert-learner return difference.

**Reward**    To bound expert-learner state distribution divergences, prior approaches have focused on sparse indicator function rewards (Ciosek, 2022), or adversarial reward learning (Ghasemipour et al., 2019). We propose a new formulation, which rewards the agent for matching the expert latent state-action pairs over an episode. In particular, for an arbitrary distance function $d$, agent state-action latent $z_t^\pi$, and a set of expert state-action latents $\mathcal{D}_E$:

$$r_t^{int}(z_t^\pi) = 1 - \min_{z^E \in \mathcal{D}_E} d(z_t^\pi, z^E) \quad (7)$$

Any function of this form rewards matching the agent's state-action pairs to the expert's, as studied in Ciosek (2022). The major differences in our formulation are that we calculate the reward on the learned model latent states, as well as compute a simple smoothed divergence, meaning an exact match isn't required for a reward. Proof A in the supplementary material shows how to make this relaxed reward compatible with the theoretical results from Ciosek (2022), such that an exact divergence bound is obtained. In particular, we prove that maximizing this reward bounds the total variation in latent-state distributions between the expert and learner, as well as bounding their extrinsic reward difference.

Intuitively, matching latent states between the learner and expert is easier than matching observations, since the representations learned from generative world model training should provide a much richer signal of state similarity. In practice, the minimization over $\mathcal{D}_E$ can be computationally expensive, so we modify the objective 7 to exactly match learner latent states to expert latents from the same time-step, as shown in Figure 1. In particular, we randomly sample consecutive expert latents $z_{t:t+H}^E$ from $\mathcal{D}_E$ and unroll the agent from the same starting state in the world model, yielding a sequence of agent latents $z_{t:t+H}^\pi$. Finally, we compute a reward at each step $t$ as follows:

$$r_t^{int}(z_t^E, z_t^\pi) = 1 - d(z_t^E, z_t^\pi) = \frac{z_t^E \cdot z_t^\pi}{\max(\|z_t^E\|, \|z_t^\pi\|)^2} \quad (8)$$

This formulation changes our method from distribution matching to mode seeking, since states frequently visited by the expert will receive greater reward in expectation. We found that this modified dot product reward empirically outperformed $L_2$ and cosine-similarity metrics.

---

**Algorithm 1** Dream Imitation (DITTO)

---

1: **Require** demonstration data $\mathcal{D} = \left\{ (x_t, a_t, x_{t+1})_{t=0}^{\|e_n\|} \mid n \in N \right\}$
2: Initialize world model parameters $\phi$
3: **while** *not converged* **do**  ▷ World model learning
4:   Draw $B_{wm}$ transition sequences $\{(x_t, a_t, x_{t+1})_{t=k}^{k+L}\} \sim \mathcal{D}$
5:   Compute all sequential RSSM components according to eqn 10
6:   Update $\phi$ with ELBO loss 11
7: **end while**
8: Initialize actor and critic parameters $\theta$, $\psi$
9: **while** *not converged* **do**  ▷ Agent training
10:   Draw $B_{ac}$ expert latent state sequences $(\hat{s}_\tau^E) \sim \hat{\mathcal{D}}^E$
11:   Generate trajectories $(\hat{s}_\tau^\pi, a_\tau)_{\tau=t}^{t+H}$ with $a_\tau \sim \pi_\theta(\cdot \mid \hat{s}_\tau)$
12:   Compute rewards $r_\tau^{\text{int}}(\hat{s}_\tau^\pi, \hat{s}_\tau^E)$ and values $v_\psi(\hat{s}_\tau^\pi)$
13:   Compute $\lambda$-returns $V_\tau^\lambda = r_t + \gamma\left((1-\lambda)v(\hat{s}_{\tau+1}^\pi) + \lambda V_{\tau+1}^\lambda\right), \quad V_{\tau+H}^\lambda = v(\hat{s}_{\tau+H}^\pi)$
14:   Update critic on $\lambda$-targets: $\sum_{\tau=t}^{t+H} \frac{1}{2}(v_\psi(\hat{s}_\tau^\pi) - sg(V_\tau^\lambda))^2$
15:   Update actor with eqn 16
16: **end while**

---

## 3.2 Algorithm

**Dataset** World model training can be performed using datasets generated by policies of any quality, since the model only predicts transition dynamics. The transition dataset is composed of $N$ episodes $e_n$ of sequences of observations $x_t$, actions $a_t$: $\mathcal{D} = \{(x_t, a_t)_{t=0}^{\|e_n\|} \mid n \in N\}$.

**World model architecture** We adapt the architecture proposed by Hafner et al. (2021), which is composed of an image encoder, a recurrent state-space model (RSSM) which learns the transition dynamics, and a decoder which reconstructs observations from the compact latent states. The encoder uses a convolutional neural network (CNN) to produce representations, while the decoder is a transposed CNN. The RSSM predicts a sequence of length $T$ deterministic recurrent states $(h_t)_{t=0}^T$, each of which are used to parameterize two distributions over stochastic hidden states. The stochastic posterior state $z_t$ is a function of the current observation $x_t$ and recurrent state $h_t$, while the stochastic prior state $\hat{z}_t$ is trained to match the posterior without access to the current observation. The current observation is reconstructed from the full model state, which is the concatenation of the deterministic and stochastic states $\hat{s}_t = (h_t, z_t)$. For further details, see the model architecture section in the appendix.

**Agent architecture** We use a standard stochastic actor-critic architecture with an entropy bonus. The actor observes Markovian recurrent states from the world model, and produces distributions over its action space, which we sample from to get actions. The critic regresses the $\lambda$-target (Sutton & Barto, 2005), computed from the sum of intrinsic rewards with a value bootstrap at the episode horizon. For further details, see the agent architecture section in the appendix.

**Algorithm** Learning proceeds in two phases: First, we train the WM on all available demonstration data using the ELBO objective 11. Next, we encode expert demonstrations into the world model latent space, and use the on-policy actor critic algorithm described above to optimize the intrinsic reward 8, which measures the divergence between agent and expert over time in latent space. In principle, any on-policy RL algorithm could be used in place of actor-critic. We describe the full procedure in Algorithm 1.

## 4 Experiments

To the best of our knowledge, we are the first to consistently recover expert performance in the pixel-based environments we study in the offline setting. Prior works generally focus on improving behavior cloning (Sasaki & Yamashina, 2021), or study a mixed setting with some online interactions allowed (Rafailov et al., 2021) (Kidambi et al., 2021). To demonstrate the effectiveness of world models for imitation learning, we train without any interaction with the true environment, nor any reward information.

Recent state-of-the-art imitation learning algorithms (Sasaki & Yamashina, 2021) (Kim et al., 2022) (Kostrikov et al., 2019) have mostly been limited in evaluation to low-dimensional perfect state observation environments. To test the effectiveness of world models for policy learning that can scale to partially-observable, high-dimensional observation environments, such as robotic manipulation from video feeds, we evaluate on difficult pixel-based environments. We test in standard pixel-based Atari environments considered by recent SOTA online methods, e.g. Brantley et al. (2020) (Reddy et al., 2020). We evaluate on a subset of the Atari domain for which strong baseline experts are available from the RL Baselines Zoo repository (Raffin, 2020), as well as a pixel-based continuous control environment.

### 4.1 Agents

To test the performance of our algorithm, we compare DITTO to a standard baseline method, behavior cloning, and to two methods which we introduce in the world model setting.

**Behavior cloning**    We train a BC model end-to-end from pixels, using a convolutional neural network architecture. Compared to prior works which study behavior cloning from pixels in Atari games (Hester et al., 2017)(Zhang et al., 2020)(Kanervisto et al., 2020), our baseline implementation achieves stronger results, even in games where it is trained with lower-scoring data.

**Dream agents**    We adapt GAIL (Ho & Ermon, 2016) and BC to the world model setting, which we dub D-GAIL and D-BC respectively. D-GAIL and D-BC both receive world model latent states instead of pixel observations. The D-BC agent is trained with maximum-likelihood estimation on the expert demonstrations in latent space, with an additional entropy regularization term which we found stabilized learning:

$$L_{BC} = \mathbb{E}_{(\hat{s},a)\sim\hat{\mathcal{D}}^E}\left[-\log\left(\pi(a|\hat{s})\right) - \eta_{BC}H(\pi(\hat{s}))\right] \tag{9}$$

The D-GAIL agent is trained on-policy in the world model using the adversarial objective from Equation 3. The D-GAIL agent optimizes its learned adversarial reward with the same actor-critic formulation used by DITTO, described in Section 3.2. D-GAIL is essentially identical to VMAIL, proposed by Rafailov et al. (2021), except that we use the world model of Dreamerv2 (Hafner et al., 2021). We train both DITTO and D-GAIL with a fixed horizon of $H = 15$. At test-time, the model-based agent policies are composed with the world model encoder and RSSM to convert high-dimensional observations into latent representations.

All model-based policies in our experiments use an identical multi-layer perceptron (MLP) architecture for fair comparison in terms of the policies' representation capacity, while the BC agent is parameterized by a stacked CNN and MLP architecture which mirrors the world model encoder plus agent policy. We found that D-GAIL was far more stable than expected, since prior works (Reddy et al., 2020) (Brantley et al., 2020) reported negative results training GAIL from pixels in the easier online setting. This suggests that world models may be beneficial for representation learning even in the online case, and that other online algorithms could be improved with world model pre-training, followed by policy training in the latent space.

We evaluate our algorithm and baselines on 5 Atari environments, and one continuous control environment, using strong PPO agents (Schulman et al., 2017) from the RL Baselines3 Zoo (Raffin, 2020) as expert demonstrators, using $N^E = \{4, 8, 15, 30, 60, 125, 250, 500, 1000\}$ expert episodes to train the agent policies in the world model. To train the world models, we generate 1000 episodes from a pre-trained policy, either PPO or advantage actor-critic (A2C) (Mnih et al., 2016), which achieves substantially lower reward compared to PPO. Surprisingly, we found that the A2C and PPO-trained world models performed similarly, and that only the quality of the imitation episodes affected final performance. We hypothesize that this is because the A2C and PPO-generated datasets provide similar coverage of the environment. It appears that the

world model can learn environment dynamics from broad classes of datasets as long as they cover the state distribution well. The data-generating policy's quality is relevant for imitation learning, but appears not to be for dynamics learning, apart from coverage.

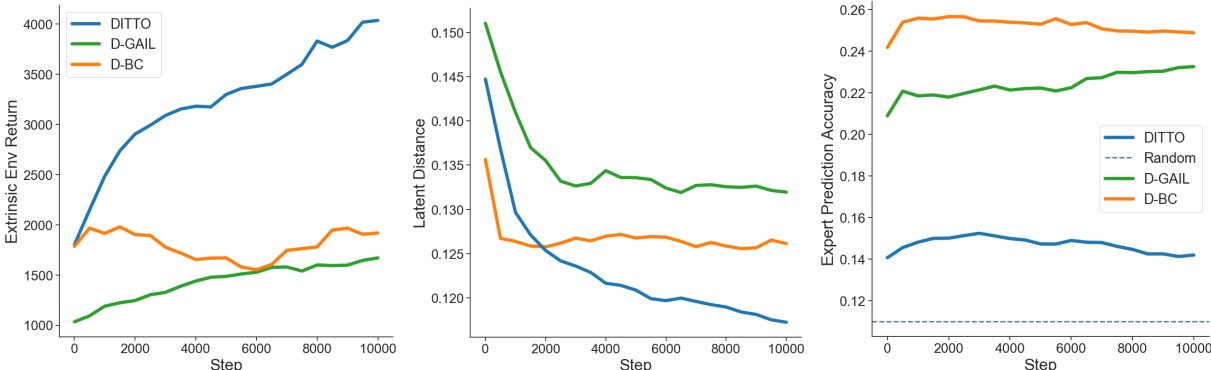

Figure 2: We compare mean extrinsic reward from rollouts in the true environment (BeamRider) throughout agent training (**left**) to agents' mean *latent distance* from the expert (**center**), and mean expert action prediction accuracy (**right**). Both latent distance and accuracy are calculated on held-out expert trajectories used for validation. Latent distance is defined as $L_d = 1 - r_{int}$. DITTO explicitly minimizes this quantity, and achieves the greatest generalization performance in the true environment. Perfect agreement with the expert would result in $L_d = 0$, but this is impossible to achieve since the world model is stochastic. Counter-intuitively, expert action prediction accuracy is *negatively* correlated with generalization performance in the true environment.

## 4.2    Results

We are interested in pushing imitation learning towards real-world deployments, which necessitate dealing with high-dimensional observations and offline learning, as mentioned in section 1. Estimating out-of-distribution imitation performance is particularly difficult in the offline setting, since by definition we do not have expert data there and cannot compare what our agent does to what an expert *would have done*. This highlights a flaw with standard offline imitation metrics such as expert action prediction accuracy, which only tell us about the learner's performance in the expert's distribution, and may not be predictive of the learner's performance under its own induced distribution.

Figure 2 shows the performance of different algorithms throughout training in the true environment, contrasted with two imitation metrics: latent distance, which we propose as a more robust measure of generalization performance for imitation; and expert action prediction accuracy, a standard imitation benchmark which is meant to capture generalization capability. DITTO achieves the lowest latent distance from expert under its own distribution in the world model. We find that counter-intuitively, *action prediction accuracy is negatively correlated with actual environment (i.e. extrinsic) performance*, whereas our latent distance measure *is* predictive of performance in the environment. This supports our hypothesis that metrics which are limited to evaluation in the expert distribution are inadequate for predicting the performance of imitation learners when deployed to the true environment, since they neglect the sequential nature of decision problems and the subsequent policy-induced covariate shift. Our results suggest that action prediction accuracy in the expert's distribution does not measure generalization performance.

Figure 3 plots the performance of DITTO against our proposed world model baselines and standard BC. In MsPacman and Qbert, most methods recover expert performance with the least amount of data we tested, and are tightly clustered, suggesting these environments are easier to learn good policies in, even with little data. D-GAIL exhibited adversarial collapse twice in MsPacman, an improvement over standard GAIL, which exhibits adversarial collapse uniformly in prior works which study imitation learning from pixels in Atari (Reddy et al., 2020)(Brantley et al., 2020). In contrast, DITTO always recovers or exceeds average

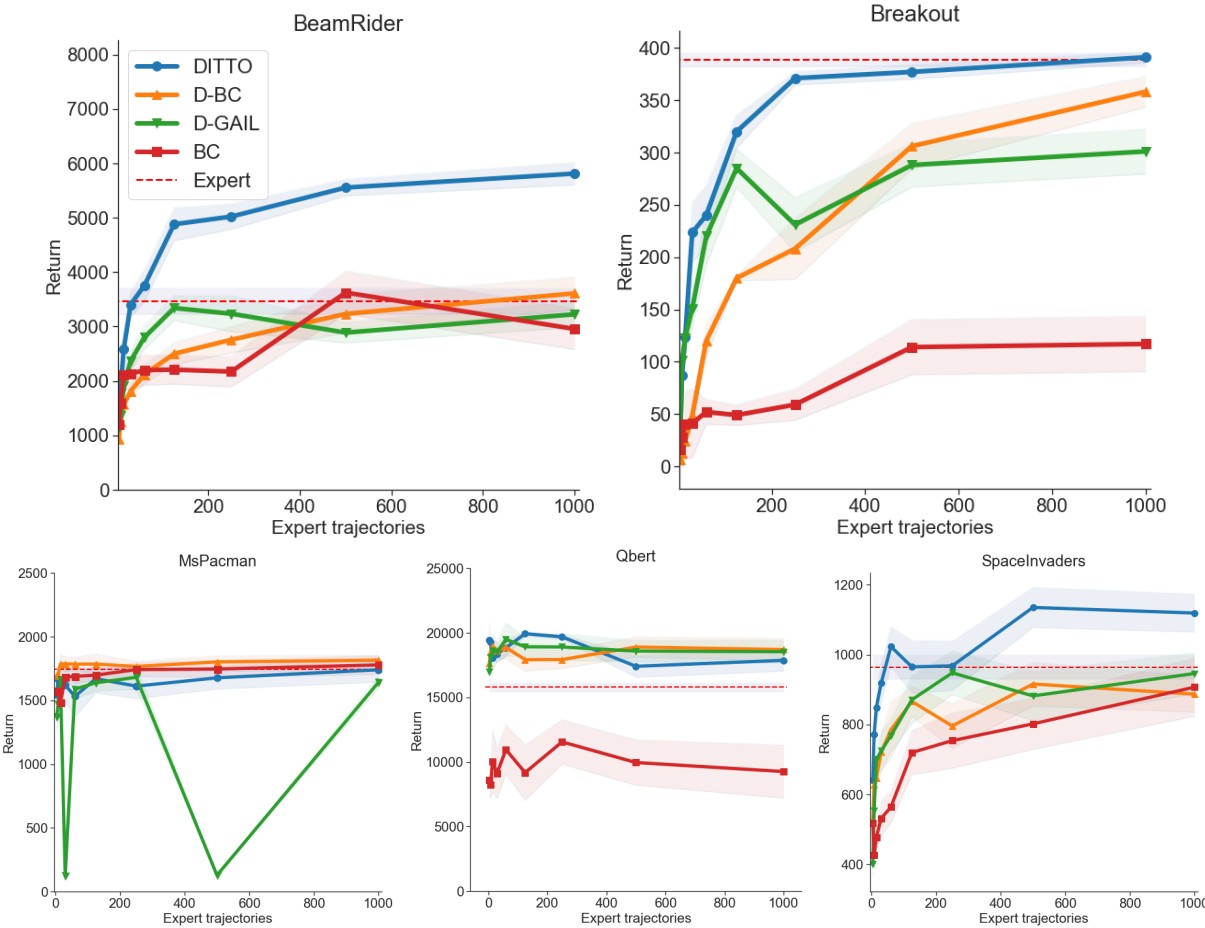

Figure 3: Results on five Atari environments from pixels, with fixed horizon $H = 15$. In all environments, DITTO matches or exceeds expert performance, and matches or exceeds all baselines. In MsPacman and Qbert, all model-based methods immediately recover expert performance with minimal data. In MsPacman, we observe adversarial collapse of D-GAIL. We follow Agarwal et al. (2021) for offline policy evaluation, and report the mean reward achieved across 10 gradient steps with 20 validation simulations, to avoid lottery-ticket policy results. Shaded regions show ±1 standard error. The experts are strong pre-trained PPO agents from the RL Baselines3 Zoo.

expert performance in all tested environments, and matches or outperforms the baselines in terms of both sample efficiency and asymptotic performance. Further results and ablations can be found in the appendix.

## 5    Conclusion

Imitation learning algorithms must deal with offline learning, high-dimensional observations spaces, and covariate shift to graduate to real-world deployment. In this work we proposed DITTO, an algorithm which addresses these problems using world models. DITTO achieves greater performance, and superior sample efficiency compared to strong baselines which we introduce for the offline setting. DITTO is the first offline imitation learning algorithm to solve these difficult Atari environments from pixels. Model-based methods are typically thought to cause generalization challenges, since agents trained in a learned model can learn to exploit generalization failures of both the dynamics or learned reward function. In contrast, our formulation encourages learners to return to the data distribution using a simple fixed reward function defined in the model latent space. By learning under their own distribution, DITTO policies mitigate policy-induced

covariate shift. Addressing the combined difficulties of high-dimensional partially observable environments and offline learning are key challenges to scale imitation learning to real world challenges.

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

# A  Proof of divergence reward bound

We prove a corollary of proposition 1 from Ciosek (2022). Ciosek (2022) uses many intermediate results and definitions, so we encourage the reader to reference their work while reading to understand this proof.

**Corollary A.1.** *Suppose we also have another imitation learner, which uses the same data-set of size N, and still satisfies Assumption 3, but instead trains on some other intrinsic reward, $R'_{int}$ which satisfies (for some $\epsilon > 0$):*

$$R'_{int}(s, a) = 1, \forall (s, a) \in D$$
$$0 \leq R'_{int}(s, a) \leq 1 - \epsilon, otherwise$$

*Let $\rho^J$ be the limiting state-action distribution of this imitation learner. Then:*

$$||\rho^J - \rho^E||_{TV} \leq \frac{\eta}{\epsilon}$$

$$\mathbb{E}_{\rho^J}[R] \geq \mathbb{E}_{\rho^E}[R] - \frac{\eta}{\epsilon}$$

*Proof.* Lemma 5 trivially still holds with $R'_{\text{int}}$ instead of $R_{\text{int}}$, as $R'_{\text{int}} \geq R_{\text{int}}$ always, $\forall \rho, \mathbb{E}_\rho[R'_{\text{int}}] \geq \mathbb{E}_\rho[R_{\text{int}}]$. Hence the bound holding true for $\mathbb{E}_{\rho^I}[R_{\text{int}}]$ implies it holds for $\mathbb{E}_{\rho^I}[R'_{\text{int}}]$ too.

Lemma 7 holds with $\kappa$ replaced by $\frac{\kappa}{\epsilon}$, so the result is $\mathbb{E}_{\rho^I}[R] \geq (1 - \frac{\kappa}{\epsilon})\mathbb{E}_{\rho^E}[R] - 4\tau_{\text{mix}}\frac{\kappa}{\epsilon}$. We do this by considering their proof in Appendix D. The properties of the intrinsic reward are utilised in just one paragraph, after equation 25. This is done in stating that $\sum_\ell \frac{\ell M_\ell}{T} \to \mathbb{E}_{\rho^I}[R_{\text{int}}]$ and $\frac{B+1}{T} \to 1 - \mathbb{E}_{\rho^I}[R_{\text{int}}]$. This is not true for $R'_{\text{int}}$. Let $p_a$ be the limiting chance of the expert agreeing with the $R'_{\text{int}}$ imitation agent. Almost by definition, $\sum_\ell \frac{\ell M_\ell}{T} \to p_a$ and $\frac{B+1}{T} \to 1 - p_a$.

Note that $\mathbb{E}_{\rho^I}[R'_{\text{int}}] \leq p_a + (1 - p_a)(1 - \epsilon)$; we yield a reward of 1 every time we agree, and at most $1 - \epsilon$ if we disagree. Hence, using $1 - \kappa = \mathbb{E}_{\rho^I}[R'_{\text{int}}]$, we have $1 - \kappa \leq p_a + (1 - p_a)(1 - \epsilon) = 1 - \epsilon + p_a\epsilon$, hence $p_a \geq 1 - \frac{\kappa}{\epsilon}$.

So, taking limits as done in the original proof, we have:

$$\mathbb{E}_{\rho^I}[R] \geq p_a\mathbb{E}_{\rho^E}[R] - (1 - p_a)4\tau_{\text{mix}} - 0$$
$$= p_a \geq p_a(\mathbb{E}_{\rho^E}[R] + 4\tau_{\text{mix}}) - 4\tau_{\text{mix}}$$
$$\geq (1 - \frac{\kappa}{\epsilon})(\mathbb{E}_{\rho^E}[R] + 4\tau_{\text{mix}}) - 4\tau_{\text{mix}}$$

Now, combining these lemmas is exactly as in section 4.4 in Ciosek (2022). The factor of $\frac{1}{\epsilon}$ carries forward, yielding $\mathbb{E}_{\rho^J}[R] \geq \mathbb{E}_{\rho^E}[R] - \frac{\eta}{\epsilon}$ as required. $\square$

# B  World Model Architecture

We adapt the recurrent state space model (RSSM) introduced by Hafner et al. (2021). The RSSM components are:

$$
\begin{array}{lll}
\text{Model state:} & \hat{s}_t = (h_t, z_t) & \\
\text{Recurrent state:} & h_t = f_\phi(\hat{s}_{t-1}, a_{t-1}) & \\
\text{Prior predictor:} & \hat{z}_t \sim p_\phi(\hat{z}_t \mid h_t) & (10) \\
\text{Posterior predictor:} & z_t \sim q_\phi(z_t \mid h_t, x_t) & \\
\text{Image reconstruction:} & \hat{x}_t \sim p_\phi(\hat{x}_t \mid \hat{s}_t) &
\end{array}
$$

All components are implemented as neural networks, with a combined parameter vector $\phi$. Since the prior model predicts the current model state using only the previous action and recurrent state, without using

the current observation, we can use it to learn behaviors without access to observations or decoding back into observation space. The prior and posterior models predict categorical distributions which are optimized with straight-through gradient estimation (Bengio et al., 2013). All components of the model are trained jointly with a modified ELBO objective:

$$\min_{\phi} \mathbb{E}_{q_\phi(z_{1:T}|a_{1:T},x_{1:T})} \left[ \sum_{t=1}^{T} -\log p_\phi(x_t \mid \hat{s}_t) + \beta D_{\text{KL-B}}(q_\phi(z_t \mid \hat{s}_t) \parallel p_\phi(\hat{z}_t \mid h_t)) \right] \tag{11}$$

where $D_{\text{KL-B}}(q \parallel p)$ denotes KL balancing (Hafner et al., 2021), which is used to control the regularization of prior and posterior towards each other with a parameter $\delta$,

$$D_{\text{KL-B}}(q \parallel p) = \delta \underbrace{D_{\text{KL}}(q \parallel sg(p))}_{\text{posterior regularizer}} + (1 - \delta) \underbrace{D_{\text{KL}}(sg(q) \parallel p)}_{\text{prior regularizer}} \tag{12}$$

and $sg(\cdot)$ is the stop gradient operator. The idea behind KL balancing is that the prior and posterior should not be regularized at the same rate: the prior should update more quickly towards the posterior, which encodes strictly more information.

## C Agent Architecture

The agent is composed of a stochastic actor which samples actions from a learned policy with parameter vector $\theta$, and a deterministic critic which predicts the expected discounted sum of future rewards the actor will achieve from the current state with parameter vector $\psi$. Both the actor and critic condition only on the current model state $\hat{s}_t$, which is Markovian:

$$\begin{aligned} \text{Actor:} \quad & a_t \sim \pi_\theta(a_t \mid \hat{s}_t) \\ \text{Critic:} \quad & v_\psi(\hat{s}_t) \approx \mathbb{E}_{\pi_\theta, p_\phi}[\Sigma_{t=0}^{H} \gamma^t r_t] \end{aligned} \tag{13}$$

We train the critic to regress the $\lambda$-target (Sutton & Barto, 2005)

$$V_t^\lambda = r_t + \gamma \left( (1 - \lambda) v_\psi(\hat{s}_{t+1}) + \lambda V_{t+1}^\lambda \right), \quad V_{t+H}^\lambda = v_\psi(\hat{s}_{t+H}) \tag{14}$$

which lets us control the temporal-difference (TD) learning horizon with the hyperparameter $\lambda$. Setting $\lambda = 0$ recovers 1-step TD learning, while $\lambda = 1$ recovers unbiased Monte Carlo returns, and intermediate values represent an exponentially weighted sum of n-step returns. In practice we use $\lambda = 0.95$. To train the critic, we regress the $\lambda$-target directly with the objective:

$$\min_{\psi} \mathbb{E}_{\pi_\theta, p_\phi} \left[ \sum_{t=1}^{H-1} \frac{1}{2} (v_\psi(\hat{s}_t) - sg(V_t^\lambda))^2 \right] \tag{15}$$

There is no loss on the last time step since the target equals the critic there. We follow Mnih et al. (2015), who suggest using a copy of the critic which updates its weights slowly, called the target network, to provide the value bootstrap targets.

The actor is trained to maximize the discounted sum of rewards predicted by the critic. We train the actor to maximize the same $\lambda$-target as the critic, and add an entropy regularization term to encourage exploration and prevent policy collapse. We optimize the actor using REINFORCE gradients (Williams, 2004) and subtract the critic value predictions from the $\lambda$-targets for variance reduction. The full actor loss function is:

$$\mathcal{L}(\theta) = \mathbb{E}_{\pi_\theta, p_\phi} \left[ \sum_{t=1}^{H-1} \underbrace{-\log \pi_\theta(a_t \mid \hat{s}_t) sg(V_t^\lambda - v_\psi(\hat{s}_t))}_{\text{reinforce}} - \underbrace{\eta H(\pi_\theta(\hat{s}_t))}_{\text{entropy regularizer}} \right] \tag{16}$$

## D Hyperparameters

## E Additional Results

Table 1: Experimental hyperparameters

| Description | Symbol | Value |
|---|---|---|
| Number of world model training episodes | $N$ | 1000 |
| Number of expert training episodes | $N^E$ | $\{4, 8, 15, 30, 60, 125, 250, 500, 1000\}$ |
| World model training batch size | $B_{wm}$ | 50 |
| World model training sequence length | $L$ | 50 |
| Agent training batch size | $B_{ac}$ | 512 |
| Agent training horizon | $H$ | 15 |
| Discount factor | $\gamma$ | 0.95 |
| $TD(\lambda)$ parameter | $\lambda$ | 0.95 |
| KL-Balancing weight | $\beta$ | 0.1 |
| KL-Balancing trade-off parameter | $\delta$ | 0.8 |
| Actor-critic entropy weight | $\eta$ | $5 \times 10^{-2}$ |
| Behavior cloning entropy weight | $\eta_{BC}$ | 0.1 |
| Optimizer | - | Adam |
| All learning rates | - | $3 \times 10^{-4}$ |
| Actor-critic target network update rate | - | 100 steps |

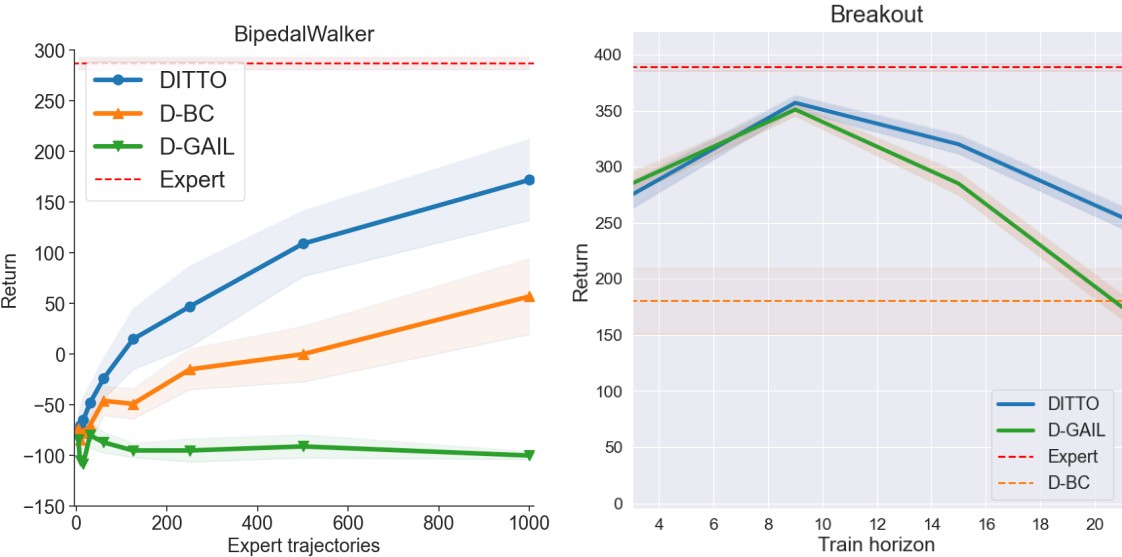

Figure 4: **Left:** Results on continuous control environment BipedalWalker, from pixels. **Right:** Training time horizon ablation. Note that both DITTO and D-GAIL achieve their maximum performance at a similar training time horizon. We conjecture that this hyperparameter is environment-specific, and report results for all environments with fixed $H = 15$.

