# OpenReview forum: "DITTO: Offline Imitation Learning with World Models"
_TMLR — Rejected by TMLR_

### Review · Reviewer_1cf7 · 2025-01-28

**Summary Of Contributions:**

This work proposes and evaluates a novel model-based method for offline imitation learning (DITTO). The core idea is that of learning a Dreamer-style dynamics model on offline data, and training an imitation policy by minimizing an IL criterion on imagined rollouts. In particular, this criterion depends on the cosine similarity between latent embeddings of states in expert and imagined trajectories. Authors argue that this solution scales to high-dimensional observations, and tackles issues related to offline learning and covariate shift. The algorithm is evaluated on a set of 5 ATARI environments, as well as on a continuous control task in the Appendix. Baselines include simple offline approaches (BC), as well as model-based variants of online and offline IL algorithms (BC and GAIL). The proposed method matches or outperforms the considered baselines.

**Audience:**

Yes

**Broader Impact Concerns:**

None.

**Claims And Evidence:**

No

**Requested Changes:**

**Changes necessary to secure this work:**
- Extend the baseline selection to include strong modern offline IL methods [2, 3]. This is fundamental for substantiating the claim of SoTA performance for imitating experts in ATARI. Furthermore, these works should be mentioned and described in the introduction and related works.
- Extend Figure 2 to the remaining ATARI environments, as a more thorough evaluation would help support claims of correlation beyond a single environment. Consider presenting results through correlation plots: select several policies trained with BC (different seeds, different number of gradient steps) and plot their environment returns against expert prediction accuracy on a validation set, and latent distance. Optionally, evaluate correlation coefficients.

**Minor changes:**
- Explain the selection of 5 ATARI environments, or extend it to include a more representative set.
- Report the necessary assumptions from [1] in the Section 3.

**Nitpicks:**
- Intro, line 7: "and the agent makes no mistakes" might need to be changed to "or the agent makes no mistakes".
- Consider extending related works to incorporate publications from 2023 and 2024, which are currently absent.
- Add numbering for equations.
- Fix broken references in Algorithm 1.

**References:**

[1] Ciosek, Imitation learning by reinforcement learning. ICLR 2022

[2] Ma et al., Versatile Offline Imitation from Observations and Examples via Regularized State-Occupancy Matching. ICML 2022

[3] Garg et al., IQ-Learn: Inverse soft-Q Learning for Imitation. NeurIPS 2021

[4] Luo et al., Optimal Transport for Offline Imitation Learning. ICLR 2023

[5] Rafailov et al., Model-Based Adversarial Imitation Learning As Online Fine-Tuning

**Strengths And Weaknesses:**

**Strengths:**
- The paper is overall well presented. Introduction and related work involve a self-contained introduction to IL. Furthermore, while the method is empirical in nature, Section 3.1 presents existing theoretical results supporting the method.
- Experiments are detailed, and empirical performance of DITTO on reported environments is convincing.

**Weaknesses:**
- The main weaknesses of this work is the lack of acknowledgement and comparison to any of the existing algorithms for offline IL beyond BC. The first paragraph in Section 4 states that "Prior works generally focus on improving behavior cloning, or study a mixed setting with some online interactions allowed [...]". However, several offline IL algorithms have been proposed, in particular in the last few years (e.g., [2, 3, 4, 5]). As the authors claim to be the "[...] first to consistently recover expert performance in the pixel-based environments we study in the offline setting.", a comparison to strong offline IL methods is necessary to support this statement. Additionally, while the discussion of previous work is nicely written, it also lacks references to more recent works in offline IL, which would need to be integrated.
- Empirical evidence for supporting correlation between latent distance and evaluation return is presented in an unusual way and not comprehensive. The analysis in itself is an interesting part of the work, but it it limited to a single environment, and it is not presented in a standard correlation plot.
- The evaluation is limited, as it mostly considers 5 ATARI environments in the main paper. How were they selected? Do the result hold for a different or larger subset of ATARI environments?
- In general, IL is a convex RL problem, and cannot always be reduced to standard, linear RL. The reduction from [1] only apply under specific assumptions, including determinism and aperiodicity (Assumptions 1,2,3 in [1]). Which of these assumptions are still needed? This should also be clearly stated in this paper. Intuitively, consider an MDP in which there is an action that never induces a change in the state. Additionally, consider the reward $r_{int}^t$ on page 5. If the agent reaches a state that perfectly matches \textit{only} one of the expert's states, there is no incentive to leave this state and match the expert's occupancy. The assumptions are crucial to exclude this case, and should therefore be reported clearly.

---

> ### Author Response · Authors · 2025-02-11
>
> Thanks for this comprehensive review!
>
> You might be surprised to hear that all of the related works you mentioned *do not actually address the setting we do*. We’ve found this confusion come up before in relation to our work. Addressing each in turn:
>
> [2] (SMODICE): uses “state” observations, not pixels.It’s not clear that their approach can be made to scale to pixel observations.
>
> [3] (IQ-LEARN): this one is tricky. For state-based envs like cartpole, they do purely offline learning. For the 3 pixel-based Atari envs they study, they have to augment the offline data with up to 1 million interactions with the environment. Hence, we do not expect their approach to work from purely offline, pixel-based observations.
>
> [4] (OTR): does not test on pixel-based observations, uses state only.
>
> [5] This is an interesting workshop paper! It is similar to our work, except that they learn an adversarial state-action discriminator which guides the policy learning. In practice, this will be almost identical to our D-GAIL method, which is one of the abselines we created. As we demonstrated, this can lead to collapse of the policy due to the inherent instability of adversarial methods, even though it does outperform vanilla BC.
>
> The state we are in appears to be that the community is actually confused about what problems have and have not been solved. To our knowledge, there is simply no other work studying this setting and recovering expert performance. Indeed, most IL works will study fewer and easier environments than we did in this work. That brings us to the next point…
>
> Environments: We do not have the compute capacity to run on the full atari suite from pixels. The world models take up to a week to train per environment. The full suite would take over an order of magnitude more compute than we have access to. If you have 1 or 2 additional envs in mind that you could explain why they are crucial, we would be willing to try them! The reason we chose these environments is to align with [R1], an excellent IL paper which was spotlighted at ICLR.
>
> Regarding the theory, we make the same assumptions as the IL by RL paper, which are not actually met in practice. The theoretical result is merely meant to demonstrate that the bounded latent reward is sound w.r.t. the expected return bound. It’s not true that if the agent reaches exactly an expert state, that there is no incentive to change: the returns incorporate entire state trajectories: the agent must match the entire trajectory to maximize return. This is one of the nice parts of our formulation, that it naturally incorporates multi-step rollouts into the learning update.
>
> We will update the paper to address the remainder of your comments, thanks for pointing out these errors!
>
> Thank you for your positive comments regarding the intellectual contributions and novelty of our approach. We tried to make our method as simple as possible, and we hope that this work clarifies the state of offline IL from pixels, and works as a strong baseline for future researchers to compare against. We believe the latent intrinsic reward idea will be exciting to other researchers studying imitation learning.
>
> [R1]: Brantley et al. “Disagreement-Regularized Imitation Learning.” ICLR 2020 spotlight.

---

> > ### Comment · Reviewer_1cf7 · 2025-02-12
> >
> > Thank you for your response.
> >
> > - I understand the argument for not comparing to [5], and that SMODICE/OTR/IQLearn are designed and largely benchmarked on state-based environments. However, to the best of my understanding, they can be easily adapted to the image-based setting (e.g. by pre-pending a small convolutional encoder to all networks). As they are not model-based methods, compute requirements should also be lower. The significance of the proposed method partially relies on existing methods not performing well in image-based setting, which is not demonstrated at the moment. Therefore, I would still recommend benchmarking one of SMODICE/IQLearn with an image encoder (or alternatively, find evidence in published works suggesting that their application to images is not possible).
> >
> > - As the environment selection is not arbitrary, but rather designed to align with existing work, I no longer believe it is strictly necessary to add other ATARI environments.
> >
> > - Thank you for taking my remaining suggestions on presentation in consideration. For what concerns theory, I would again suggest to clearly report the assumption you inherit from previous works.

---

> > > ### Author Response · Authors · 2025-03-10
> > >
> > > Regarding the baselines: our discussion with another reviewer above indicated that our baseline method D-GAIL is essentially identical to VMAIL [1], except that it uses the greater performing Dreamerv2 world model. We will update the text to indicate this more clearly, which should help readers contextualize the performance of DITTO relative to this strong baseline.
> > >
> > > [1]: Rafailov et al., "Visual adversarial imitation learning using variational models" (NeurIPS 2021)

---

### Review · Reviewer_b4TG · 2025-02-05

**Summary Of Contributions:**

This paper introduces an offline imitation learning method that leverages a learned world model and an intrinsic reward to align agent and expert trajectories in the model's latent space. This approach is well-motivated. Experiments in atari games demonstrate its effectiveness over baselines including vanilla behavior cloning and GAIL.

**Audience:**

Yes

**Broader Impact Concerns:**

No broader impact concern

**Claims And Evidence:**

No

**Requested Changes:**

Additional experiments:
- test in more complex control tasks with continuous action space
- swap the backbone/baseline BC with diffusion policy

Format: There are no equation numbers but equations and losses are being referred in text and in Algorithm 1

**Strengths And Weaknesses:**

The idea is well motivated and the idea to combine world models and intrinsic rewards is novel.
The paper is generally easy to follow (except the formatting issues mentioned below).

The major weakness is the empirical results are not sufficient to fully support the claims. First, the experiments are in simple atari domains with discrete action spaces. At the same time, it is unclear whether the real problem here is multimodal demonstration data or truly the covariate shift as the paper hypothesis. With multimodal data, BC loss has an averaging effect that drive the learned policy out-of-distribution. The intrinsic reward in DITTO may have an effect on favoring one mode and therefore allows the policy to commit to one action. An easy way to decouple this is to use more powerful backbone that can model the action distribution, such as diffusion policy [1].

I suggest the authors to add additional experiments and thorough ablations to improve this work.


[1] Chi, C., Xu, Z., Feng, S., Cousineau, E., Du, Y., Burchfiel, B., ... & Song, S. (2023). Diffusion policy: Visuomotor policy learning via action diffusion. The International Journal of Robotics Research, 02783649241273668.

---

> ### Author Response · Authors · 2025-02-11
>
> Thanks for your review!
>
> We are not particularly focused on the continuous action setting in this work, which is the primary setting for diffusion policies. There is no issue with representing multi-modal discrete action distributions with standard methods like behavior cloning.
>
> The focus of this work is to investigate whether practicing rollouts in a world model can induce imitation learning--not on the particular details of the policy learning architecture, which is orthogonal. The fact that we use actor-critic RL is simply to demonstrate the idea using the simplest possible baseline. The work we consider closest to ours in spirit is [1], which studies a novel IL formulation in the same Atari environments we study, which is why we chose these. We also included the most difficult continuous control task they test on, BipedalWalkerv2, in the Appendix; though we make it even more challenging by learning from pixels alone.
>
> This again points to the main contribution of our work: can world models learned only from pixel observations provide good enough latent states, and latent state dynamics, for imitation learning in the world model to succeed, and transfer back to the original environment? We answer this question strongly in the affirmative with our formulation. While more ablations and comparisons are always good, we believe this would distract from the central point of our work, which is to demonstrate how multi-step IL via RL in the world model latent space recovers expert performance. We are excited to see how others build on these ideas and integrate them with other advances in policy learning, such as the diffusion policy work.
>
> Thanks for pointing out the formatting issues, we will update this!
>
> [1]: Brantley et al. “Disagreement-Regularized Imitation Learning.” ICLR 2020 spotlight.

---

### Review · Reviewer_ddhp · 2025-02-13

**Summary Of Contributions:**

This paper introduces DITTO a Model-Based Imitation Learning algorithm from pixels. Ditto is trained entirely on the latent space learned by a recurrent state space model by using a similarity measure as intrinsic reward.

**Audience:**

Yes

**Claims And Evidence:**

No

**Requested Changes:**

## Related work:

Several relevant papers on model-based imitation learning from pixels and RSSM-based approaches are missing, including:

    [1] Hu et al., "Model-based imitation learning for urban driving" (NeurIPS 2022)
    [2] Rafailov et al., "Visual adversarial imitation learning using variational models" (NeurIPS 2021)
    [3] Rafailov et al., "Offline reinforcement learning from images with latent space models" (L4DC 2021)
    [4] Kolev et al., "Efficient Imitation Learning with Conservative World Models" (arXiv 2024)

Similarly, literature on imitation from videos without expert actions, such as:

    [5] Giammarino et al., "Adversarial Imitation Learning from Visual Observations using Latent Information" (TMLR)
    [6] Liu et al., "Visual Imitation Learning with Patch Rewards" (ICLR 2023)

should be considered to properly position this work. A comparison with VMAIL [2] and MILE [1] is particularly relevant. If a direct comparison is not feasible, explaining why would clarify the contributions of DITTO.

## Experiments
The evaluation is limited in terms of tested environments and baselines. Broader benchmarking, such as DeepMind Control Suite or Adroit, would strengthen the results. If these environments are unsuitable, stating why would provide clarity.

Similarly, VMAIL [2] appears closely related to DITTO. If it does not represent a fair baseline, discussing the reasons would improve the positioning of the work.

## Ablation Studies and Limitations
The method relies on a well-trained world model, yet challenges in achieving this, as noted in [5], are not addressed.

Adding ablations would clarify the method’s strengths and weaknesses. Additionally, discussing limitations (e.g., failure cases) would improve transparency.

##  Theoretical Results
The theoretical contributions could be included in the main paper for accessibility, while the appendix could detail intermediate results from Ciosek (2022) for completeness.

## Novelty
The lack of key citations and experimental comparisons makes it difficult to assess the novelty of DITTO. Clarifying the unique contributions relative to existing methods would improve the positioning of this work.

**Strengths And Weaknesses:**

Strengths:
1. The paper is clear and the algorithm is well-presented and easy to understand.
2. The experiments show promising results compared to the tested baselines.

Weaknesses:
1. Related Work: the Related Work section does not mention many relevant studies on Model-Based Imitation Learning and Imitation Learning from pixels.
2. Limited number of experiments and limited baselines comparison.
3. Absence of ablation studies.
4. Absence of a limitations section.
4. Theoretical result in the Supplementary materials.

---

> ### Author Response · Authors · 2025-02-27
>
> Thanks for this clear and thorough review, it’s made it clear to us how to better position our work:
>
> One approach to imitation learning is to learn to match the state distribution of the expert (as opposed to directly learning the action distribution/policy). Model-based methods improve the power of state-based approaches by producing compact and semantically meaningful latent state representations through the world model learning. The question then is how exactly to use the learned latent states to induce a good policy. The most common method is to use an adversarial approach, so that a learned discriminator tries to distinguish between the policy-generated and expert latent state trajectories. This is essentially the approach of VMAIL, as well as our baseline D-GAIL. The two algorithms are nearly identical—we simply use the more powerful world model of Dreamerv2, and do not simultaneously optimize the model and policy. Our world model is learned once offline, then rollouts in the WM are used to train the policy. We expect that D-GAIL and VMAIL should perform similarly, as the approaches are nearly identical. The primary issue with this approach is the instability of adversarial optimization, as we discussed.
>
> The central contribution of DITTO is to replace the adversarially learned discriminator with a fixed latent divergence function, which is a kind of inner product. We find that this intrinsic latent reward is more stable, learns faster, and achieves overall better performance than the adversarial approach of our baseline, D-GAIL.
>
> Another metric that is used to encourage policies to remain close to the expert state distribution is ensemble-disagreement methods like [3]. We don’t explicitly discuss [3] because it focuses on RL rather than IL, whereas the work we primarily discuss on this topic, DRIL, uses this same disagreement idea for imitation learning.
>
> Without the learned model latents, it is more difficult to use state-based matching approaches for pixel-based environments, so they must rely more on actions or other means of estimating state similarity. We believe that the set of fundamental ideas relevant to our work are world models, disagreement-based imitation, and adversarial policy optimization; hence we focused our attention on the key papers in these domains for the related work discussion.
>
> We will update the discussion in the related work to make our contribution more clear, and compare with some of the methods you mentioned, especially to point out that D-GAIL, the baseline we developed, is essentially the same as VMAIL. We will also make it more clear that these environments were chosen to align with DRIL, a well-known and well-regarded paper that studies imitation learning.
>
> The question of learning from videos without actions is interesting, but orthogonal to our focus in this work.
>
> Another reviewer has asked the we include an additional baseline method for comparison, which we are currently working on results for. We will soon post an updated revision with more clear comparison to related work as mentioned above, an the additional baseline method.

---

> ### Comment · Reviewer_ddhp · 2025-02-28
>
> Thank you for your response and for providing further clarifications.
>
> I have a few follow-up questions. Imitation learning from videos—without access to actions—can also be framed as matching the expert's state distribution. For example, adversarial imitation learning can be employed, with the discriminator defined over transitions rather than actions (i.e., using $D(s,s′)$ instead of $D(s,a)$ or $D(z,z')$ rather than $D(z,a)$). Additionally, some literature proposes techniques that involve learning a latent space without necessarily relying on a world model.
>
> I am unclear why the authors claim that this literature is orthogonal. Would the fixed latent divergence function proposed in DITTO also be applicable in this setting? Does this inner product require the use of an offline-learned world model to function effectively?
>
> On another note, how do you explain that DITTO exceeds expert performance? Since the problem is formulated as state distribution matching, the objective should be to replicate expert performance. Exceeding expert performance is, therefore, quite surprising.

---

> > ### Author Response · Authors · 2025-03-10
> >
> > Thanks for these follow-up questions.
> >
> > Because we are targeting the offline setting, we cannot get on-policy transitions without a world model. That is the main benefit of the world model: it enables on-policy learning, offline. It is true that the DITTO reward formulation doesn't rely on actions; it only compares latent states. But it is unclear how would one get on-policy transition latent states without a world model. For the online case where one can get on-policy state transitions, we imagine the DITTO reward could be used as an alternative to adversarial or other state-matching methods.
> >
> > Regarding the agent sometimes outperforming the expert: one possibility is that this formulation is mode-seeking rather than distribution matching. That is, to maximize the latent matching rewards under uncertainty (stochastic transitions in the world model), it may be that the optimal policy tends towards the mode, neglecting outlier states/actions.

---

### Decision · Action_Editor_PDdM · 2025-03-18

**Recommendation:** Reject

**Comment:**

Relating the proposed method to existing work is challenging due to the absence of state-of-the-art baselines in the specific environment and problem setting under consideration. This underexplored area of offline visual imitation learning requires significantly more thorough evaluations, which would necessitate a major revision that would need to be resubmitted as a new submission.

**Audience:**

Offline imitation learning from images is a highly relevant problem, for example for learning from online videos. The proposed method, which trains a world model from the data set in the first stage, and learns a policy in the second stage using model-based (latent) roll-outs with a cosine-similarity reward seems interesting. Hence, I argue, that the submission would be sufficiently interesting to the TMLR's audience, if additional evidence were provided.

**Claims And Evidence:**

I agree with the reviewers that the submission does not provide sufficient evidence to support the claimed performance with respect to state of the art methods. Although offline imitation learning from images is a challenging setting in general, we can not conclude that existing offline IL methods would fail just because they have not been tested in this setting. The difficulty of an imitation learning task may vary substantially depending on the quantity and quality of demonstrations, the considered MDP and the underlying task, and the difficulty of the considered discrete Atari tasks can not be established based on simple baselines like BC, GAIL and their basic model-based variants. The authors argued that the Atari benchmark were chosen as they were used in related work [R1], but [R1] was not considering the offline setting.

Hence, I agree with the general sentiment of the reviewers, that the submission lacks sufficient experimental validation and needs to be evaluated significantly more thoroughly with respect to state-of-the-art offline imitation learning methods, such as IQ-Learn. Such evaluations should make use of straightforward best practices from the "RL from images" settings, such as suitable encoder architectures, data augmentation and auxiliary representation learning tasks. Such thorough evaluation would be crucial to strengthen the claimed benefits of the proposed method.

An additional related work that seems relevant, is the work by Pari et al. 2020, which demonstrated on a real robot, that representation learning is effective for visual (offline) imitation learning.

__References__

Pari, J., Muhammad, N., Pandian, S., & Pinto, L. The Surprising Effectiveness of Representation Learning for Visual Imitation. Robotics: Science & Systems. 2020.

**Resubmission Of Major Revision:**

The authors may consider submitting a major revision at a later time.